# Animal Models of Coenzyme Q Deficiency: Mechanistic and Translational Learnings

**DOI:** 10.3390/antiox10111687

**Published:** 2021-10-26

**Authors:** Pilar González-García, Eliana Barriocanal-Casado, María Elena Díaz-Casado, Sergio López-Herrador, Agustín Hidalgo-Gutiérrez, Luis C. López

**Affiliations:** 1Departamento de Fisiología, Facultad de Medicina, Universidad de Granada, 18016 Granada, Spain; elianabc@ugr.es (E.B.-C.); elenadiaz@ugr.es (M.E.D.-C.); sergiolopezhe@correo.ugr.es (S.L.-H.); ahg@ugr.es (A.H.-G.); 2Centro de Investigación Biomédica, Instituto de Biotecnología, Universidad de Granada, 18016 Granada, Spain

**Keywords:** preclinical models, pathomechanisms, therapy, mitochondria, mitochondrial disease, bioenergetics, oxidative stress, sulfide metabolism

## Abstract

Coenzyme Q (CoQ) is a vital lipophilic molecule that is endogenously synthesized in the mitochondria of each cell. The CoQ biosynthetic pathway is complex and not completely characterized, and it involves at least thirteen catalytic and regulatory proteins. Once it is synthesized, CoQ exerts a wide variety of mitochondrial and extramitochondrial functions thank to its redox capacity and its lipophilicity. Thus, low levels of CoQ cause diseases with heterogeneous clinical symptoms, which are not always understood. The decreased levels of CoQ may be primary caused by defects in the CoQ biosynthetic pathway or secondarily associated with other diseases. In both cases, the pathomechanisms are related to the CoQ functions, although further experimental evidence is required to establish this association. The conventional treatment for CoQ deficiencies is the high doses of oral CoQ_10_ supplementation, but this therapy is not effective for some specific clinical presentations, especially in those involving the nervous system. To better understand the CoQ biosynthetic pathway, the biological functions linked to CoQ and the pathomechanisms of CoQ deficiencies, and to improve the therapeutic outcomes of this syndrome, a variety of animal models have been generated and characterized in the last decade. In this review, we show all the animal models available, remarking on the most important outcomes that each model has provided. Finally, we also comment some gaps and future research directions related to CoQ metabolism and how the current and novel animal models may help in the development of future research studies.

## 1. Introduction

Coenzyme Q (CoQ) or ubiquinone is an essential lipid present in almost all living organisms. CoQ is composed of a benzoquinone ring and a polyisoprene chain of variable length. Each specie has a major CoQ form attending to the length of the polyisoprene chain, i.e., 10 isoprene units (CoQ_10_) in humans, zebrafish (*Danio rerio*), and *Schizosaccharomyces pombe*, 9 units (CoQ_9_) in mice, *Caenorhabditis elegans* and plants, 8 units (CoQ_8_) in *Escherichia coli*, and 6 units (CoQ_6_) in *Saccharomyces cerevisiae* [1,2]. Although one form of CoQ is dominant in each organism, a minor form of CoQ can be also detected in some species, e.g., CoQ_9_ in humans or CoQ_10_ in mice [3].

Thanks to its redox chemistry, CoQ participates in the mitochondrial respiration, accepting electrons from complex I or complex II and transferring them to complex III at the same time that a proton gradient is produced in the intermembrane space [4,5]. This proton motive force is used by the ATP synthase to produce the final ATP molecule. Moreover, CoQ is a cofactor for different mitochondrial dehydrogenases involved in different metabolic pathways [1,4,6], including the following: the dihydroorotate dehydrogenase (DHODH), involved in the pyrimidine biosynthesis [7,8]; the mitochondrial glycerol-3-phosphate dehydrogenase (G3PDH), linking glycolysis, oxidative phosphorylation, and fatty acid metabolism [9,10]; the electron transport flavoprotein dehydrogenase (ETFDH), involved in fatty acid β-oxidation, the catabolism of several amino acids, and sarcosine metabolism [11,12]; the proline dehydrogenase (PRODH) and proline dehydrogenase 2 (PRODH2) required for proline, glyoxylate, and arginine metabolism [6,13]; the choline dehydrogenase (CHDH) related to glycine metabolism [14]; and the sulfide-quinone oxidoreductase (SQOR) that catalyzes the first reaction required for the detoxification of hydrogen sulfide (H_2_S) [15] (Figure 1a).

Uncoupling proteins (UCPs) are other mitochondrial components susceptible to be regulated by redox reactions. Some experimental evidence has suggested that CoQ is a cofactor for UCPs, connecting CoQ with the control of thermogenesis [16,17] and with an alternative mechanism for decreasing reactive oxygen species (ROS) inside mitochondria via UCPs–superoxide interaction [18]. However, controversial results have been reported on the involvement of CoQ in the regulation of the UCPs [17,19]. Other studies investigated the regulation of the mitochondrial permeability transition pore (PTP) by ubiquinone analogues [20,21]. Experimental evidence suggests that quinones modulate the PTP through a common binding site rather than through redox reactions. CoQ seems also to prevent PTP opening [22], but further studies are required to understand whether CoQ has a primary function in the PTP. Recently, two studies reported that CoQ could also act as a cofactor for the ferroptosis suppressor protein 1 (FSP1) (previously known as apoptosis-inducing factor mitochondrial 2, AIFM2), thus contributing to the suppression of ferroptosis, an iron-dependent form of cell death with increased oxidative damage [23,24] (Figure 1b).

In addition to these metabolic roles related to mitochondria, CoQ is a crucial antioxidant that specially acts in the lipid membranes. The antioxidant capacity of CoQ is due to its capability to directly reduce reactive oxygen species (ROS) but also to regenerate other antioxidants, e.g., tocopherol and ascorbate [2,6,25,26]. Moreover, different non-mitochondrial functions of CoQ have been identified, and extensive reviews of these actions have been published elsewhere [1,27,28].

The synthesis of CoQ mainly occurs in the mitochondrial inner membrane by a set of nuclear-encoded COQ proteins through a biochemical pathway that is not completely understood [2,4,10,25,29] (Figure 2a). However, some authors have also located the synthesis of CoQ outside the mitochondria, specifically in the endoplasmic reticulum (ER)–Golgi system [30,31]. Moreover, a study in yeast demonstrated the role of the endoplasmic reticulum–mitochondria encounter structure (ERMES) in the coordination of CoQ biosynthesis in mitochondria, highlighting the importance of the communication between these two organelles [32]. The CoQ biosynthetic process in the mitochondria is similar in prokaryotes and eukaryotes: a long polyisoprenoid lipid tail is coupled to a benzenoid precursor, and the benzenoid ring is further modified through successive steps to yield the final product [2,33,34]. 4-Hydroxybenzoic acid (4-HB) is the precursor of the benzoquinone ring of CoQ [35], although some other alternative ring precursors are being postulated [2]. For the biosynthesis of 4-HB, human cells utilize phenylalanine or tyrosine as ring precursors [29]. The isoprene carbon units for making the CoQ side-chain are derived from the mevalonate pathway in eukaryotes and some prokaryotes [36]. The lipophilic polyprenyl tail is synthesized by the addition of dimethylallyl pyrophosphate (DMAPP) and isopentenyl pyrophosphate (IPP) by PDSS1/PDSS2 in humans and mice. Consequently, the heterotetramer PDSS1/PDSS2 is responsible for the length of the polyprenyl tail [29,36,37]. Then, COQ2 in mammals catalyze the attachment of the polyisoprenoid tail to the ring precursor. From this step, a total of seven reactions (one decarboxylation, three hydroxylation, and three methylation) produce the fully substituted benzoquinone ring of CoQ [35]: C5-hydroxylation (human COQ6), O-methylations (human COQ3), C1-hydroxilation and C1-decarboxylation (unidentified), C2-methylation (human COQ5), and C6-hydroxilation (human COQ7) [4,38,39,40,41]. In addition to the enzymes with catalytic activity in the CoQ biosynthetic pathway, there are additional proteins necessary for CoQ synthesis, although their exact functions are not well established. However, the disruption in any of them produce a reduction in the CoQ levels. In humans, these proteins are COQ4, COQ8A (also known as ADCK3), COQ8B (also known as ADCK4), COQ9, COQ10A, and COQ10B [42,43,44,45,46,47]. Some of the biosynthetic proteins are assembled into a high molecular mass complex, named complex Q, with the objective of improving the catalytic efficiency of the CoQ biosynthetic pathway and minimizing the escape of intermediates [4,29,37] (Figure 2b). The benzoquinone ring of CoQ can be found in its completely oxidized form (ubiquinone, CoQ), in the semireduced form (ubisemiquinone, CoQH●), and in the completely reduced form (ubiquinol, CoQH_2_) after receiving two electrons [1,4] (Figure 2c).

CoQ levels can be severely reduced in a group of mitochondrial diseases named CoQ deficiencies, which are clinically and genetically heterogeneous disorders. Five major phenotypes have been described: (1) encephalomyopathy (recurrent myoglobinuria, encephalopathy, and mitochondrial myopathy); (2) cerebellar ataxia (cerebellar atrophy associated with other neurologic manifestations and, occasionally, endocrine dysfunctions); (3) infantile multisystemic form; (4) isolated myopathy, characterized by muscle weakness, myoglobinuria, exercise intolerance, and elevated creatine kinase (CK); and (5) nephropathy. Growth retardation, deafness, hearing loss, and cardiomyopathy have also been described in CoQ_10_-deficient patients [48]. This heterogeneity in the clinical presentations suggests the existence of multiple pathomechanisms, which could be theoretically due to the multiple functional roles of CoQ and the complexity of the CoQ biosynthetic pathway, although further experimental evidences are required to understand the diseases’ conditions and tissue-specific pathomechanisms. Primary CoQ deficiency is caused by autosomal recessive mutations in CoQ genes [48], while secondary CoQ deficiency is produced by mutations in genes unrelated to CoQ biosynthesis or is derived from other physiological processes or pharmacological treatments [49,50]. Mutations or deletions in 10 genes involved in the CoQ biosynthetic pathway have been reported in humans: *PSSS1* [51], *PDSS2* [52], *COQ2* [51,53,54], *COQ4* [55], *COQ5* [56], *COQ6* [57], *COQ7* [58,59], *COQ8A* (or *ADCK3*) [60,61], *COQ8B* (or *ADCK4*) [62,63], and *COQ9* [64,65]. The secondary forms are probably more frequent, and they include patients with ataxia and oculomotor apraxia due to mutations in the aprataxin (APTX) gene [66,67,68]; with isolated myopathy due to mutations in the electron-transferring-flavoprotein dehydrogenase gene (ETFDH) [69]; and with cardiofaciocutaneous (CFC) syndrome due to mutations in BRAF [70]. Moreover, CoQ_10_ deficiency has been reported in association with pathogenic mitochondrial DNA (mtDNA) depletion, deletions, or point mutations [71,72,73]. In addition, secondary CoQ deficiency has been linked to a decrease in the levels of proteins of the complex Q in various mouse models of mitochondrial diseases [74], as well as in the muscle and adipose tissue of patients and a mouse model with insulin resistance [75]. Furthermore, the levels of CoQ are one of the mitochondrial factors involved in aging, and a specific review about this topic was recently published [29].

Supplementation with exogenous CoQ_10_ is the main therapeutic strategy for these pathologies. However, the therapeutic efficacy of CoQ_10_ supplementation is quite variable [48,50,76,77], highlighting the need to look for new options. Thus, this review is focused on describing the available animal models for CoQ deficiency that have been used to date in the study of CoQ deficiencies (Table 1), how they have contributed to expand our knowledge about CoQ metabolism and the pathomechanisms of CoQ deficiency, and how they may provide optimal therapeutic alternatives for patients suffering from CoQ deficiencies.

## 2. Invertebrate Models of CoQ Deficiency

Invertebrate models have the great advantage of being easily generated and characterized compared to vertebrate models. Thus, worms and flies lacking CoQ have been produced, and they highlighted important functions of CoQ. However, invertebrate systems are not sufficient to reproduce the complexity of mammalian systems [125,126].

### 2.1. Fruit Fly Models: Drosophila Melanogaster

*Drosophila melanogaster* is commonly known as the fruit fly or vinegar fly [126]. It has been widely used because it offers several advantages for the investigation of molecular and cellular mechanisms due to its short lifespan, large number of offspring, and having a genome that is easy to modulate [127]. *Drosophila* contains three CoQ forms with the following approximated proportions: 5% CoQ_8_, 82% CoQ_9_, and 13% CoQ_10_. Those proportions change depending on the age of the fly and the developmental stage [83].

Mutations in the *Drosophila qless* (*cg31005*) gene, an orthologue of the human *PDSS1,* induce an upregulation of markers of mitochondrial stress and caspase-dependent apoptosis in neurons. Full rescue of the *qless* neural phenotype was achieved by dietary supplementation with CoQ_4_, CoQ_9_, or CoQ_10_ [78]. Additionally, mutations in the *Drosophila sbo* (*cg9613 or coq2*) gene, homolog of the human *COQ2*, leads to a small larvae phenotype. The *sbo* null mutants are developmentally arrested at the first instar larval stage. Flies that are heterozygous for *sbo* show reduced CoQ_9_ and CoQ_10_ production and a controversial extended lifespan [79]. *Coq2* mutant flies are more susceptible to bacterial and fungal infections, while they are more resistant to viruses. The supplementation with CoQ_10_ partially rescues the impaired immune functions of *coq2* mutants because it restores the gene expression of anti-microbial genes but increases the susceptibility to viral infection [80]. Moreover, another *Drosophila* model with mutation in the *coq2* gene was generated employing garland cell nephrocytes (GCN) to model human *COQ2* nephropathy. This study found that *coq2* is required for slit diaphragm morphology and function, since *coq2* silencing causes large areas without slit diaphragms, while slit diaphragms densely populate the surface of control GCN. Following, it was proved that the pathogenesis of *coq2* is linked to mitochondrial reactive oxygen species (ROS) formation because of the increased ROS formation in *coq2* loss-of-function. According to that, the phenotype was rescued after the treatment of *coq2* flies with the ROS scavenger glutathione for 5 days, remarking the importance of oxidative stress in the renal pathology associated to CoQ deficiency. Interestingly, this *coq2* phenotype was also partially rescued by the supplementation with vanillic acid (VA) [81]. VA is an analog of the CoQ natural precursor, the 4-hydroxybenzoic acid (4-HB), which has previously demonstrated its therapeutic activity in other models of CoQ deficiency [35,128], although the therapeutic mechanism for defects in *coq2* have not been clearly identified.

The UAS-GAL4 system for gene silencing with RNAi has been also used in *Drosophila* to interfere with the expression of each of the CoQ biosynthetic genes [83]. The results demonstrate that flies with RNAi against CoQ genes show a decrease in CoQ levels. The percentage of decrease depends on the affected gene and the intensity of the gene silencing. RNAi against the gene *cg10585* (human *PDSS2*) showed the hardest phenotype with flies arresting their development cycle after egg hatching. Gene interference of *qless (cg31005)*, *coq2*, *coq3*, *coq5*, *coq7*, *coq8*, *coq9*, or *coq10* produced lethality at larvae or pupae development stages. When the silencing gene was *coq6*, flies managed to achieve the adult fly stage but suffered from severe CoQ deficiency. Demethoxyubiquinone (DMQ) was detected when *coq3*, *coq6*, *coq7*, and *coq9* were silenced, supporting the idea of a multi-enzymatic complex for CoQ biosynthesis [82,83].

### 2.2. Worm Models: Caenorhabditis Elegans

*Caenorhabditis elegans* (*C. elegans*) has been extensively used as a model organism in different fields of research [29,126]. The ability to inactivate a target gene transiently by RNAi has greatly accelerated the analysis of loss-of-function phenotypes in *C. elegans* [129].

A model of secondary CoQ deficiency in *C. elegans* is the *mev-1* mutant strain, which has a defect in complex II. This model shows a decrease in the CoQ_9_ levels and altered rates of the reduced and oxidized forms of CoQ [84,126]. *Mev-1* mutant also manifests a disruption of the superoxide dismutase activity, leading to an accumulation of reactive oxygen species (ROS) and a shorter lifespan compared to the wild type [85]. The supplementation with CoQ_10_ rescues the phenotype by the reduction of superoxide anion level, suggesting the link between lifespan and oxidative stress in mitochondria [86]. Models of primary CoQ deficiency in *C. elegans* will be described below.

*C. elegans clk-1* mutants are worms with primary CoQ deficiency. The *clk-1* gene is the homologous of the *COQ7* gene in humans. As a result of this mutation, the *clk-1* mutant shows a decrease in the CoQ_9_ levels and an accumulation of the intermediate demethoxyubiquinone (DMQ_9_), the substrate of COQ7 [87], thus confirming the catalytic reaction of this enzyme. Intriguingly, the phenotype is characterized by an increase in the lifespan together with a slowed pharyngeal pumping and abnormalities in defecation, movement, embryogenesis, and larvae development [89,130,131]. The reason for the life extension seems to be a lower production of reactive oxygen species (ROS) due to a reduction in the electron flow of the respiratory chain because of the inhibition of complex I [126]. This hypothesis is based on (1) *Clk-1* mutants showing a profound defect in complex I+III activity, which contributes to ROS production and its link to lifespan [88]; and (2) the accumulated DMQ_9_ in *clk-1* mutant could inhibit complex I+III activity [90]. Nevertheless, it is also important to take into account that either the small amount of CoQ_9_ synthesized by *clk-1* mutant and/or the CoQ_8_ coming from the diet would stabilize complex III and determine the longevity of this model [87,126,132]. The supplementation with CoQ_10_ to *clk*-1 mutant rescued the phenotype and returned the lifespan to wild-type levels [91,126]. Other studies in the *clk-1* mutant indicated that CoQ_10_ is the CoQ isoform with higher antioxidant properties because the clk-1 mutant feed with genetically engineered bacteria that produce CoQ_10_ rescued the phenotype by lowering oxidative stress, while those fed with bacteria that produce CoQ_6_, CoQ_7_, CoQ_8_, or CoQ_9_ had no effect [86,89,92,126].

*C. elegans* mutant strains VC479, VC752, and VC614 are available heterozygous mutant strains with deletions in the genes *coq-1* (allele ok749), *coq-2* (allele ok1066), and *coq-8* (allele ok840), respectively. *Coq-1*, *coq-2,* and *coq-8* knockouts arrested development during larval stages L1–L2 (*coq-1* mutants) or L4-adult (*coq-2* and *coq-8* mutants) [93]. Larval development arrest was also reported for *C. elegans coq-3* knockouts, pointing out the importance of CoQ in the early stages of development [95]. The first generation of *coq-1* and *coq-2* knockouts had muscular atrophy due to cell death, apoptosis corpses, and eventual tissue disorganization with paralysis of the posterior half of the larval body, which was probably mimicking the encephalomyopathic form of CoQ deficiency [93]. Other *coq-1*, *coq-2,* and *coq-3* knockdown models mimic the cerebellar involvement of CoQ deficiency with an age-dependent loss of motor coordination correlated with the progressive degeneration of GABA neurons. However, other types of neurons in motor and sensory circuits that use other neurotransmitters (dopamine, acetylcholine, glutamate, serotonin) and body muscle cells were less sensitive to CoQ depletion [94]. The supplementation with exogenous CoQ_10_ or feeding worms with bacteria containing its own CoQ_8_ did not have a great impact on phenotype improvement of *coq-1*, *coq-2,* or *coq-3* mutants, suggesting that exogenous CoQ has therapeutic limitations due to its limited bioavailability [92]. A full rescue of the *coq-3* mutant phenotype was achieved by an extra-chromosomal array containing the own *C. elegans coq-3* gene, illustrating the crucial role that the endogenous synthesized CoQ_9_ isoform plays in fertility and development [92,126]. Alternatively, *coq-8* mutants increased their lifespan if they were fed with a CoQ-rich diet with wild-type *E. coli*, but no improvement was observed in fertility [96].

## 3. Vertebrate Models of CoQ Deficiency

Vertebrate models of CoQ deficiency, especially the mammalian ones, are helpful to study the physiopathology of CoQ deficiencies and to understand the heterogeneity of these syndromes because they mimic the cellular functions of CoQ, the biosynthesis, its regulation, and the tissue specificity that may exist in humans. That is the reason why these models are also useful in preclinical studies to test new therapies [125].

### 3.1. Zebrafish Models of CoQ Deficiency

Zebrafish biology allows ready access to all developmental stages, and the optical clarity of embryos and larvae allow real-time imaging of developing pathologies. Sophisticated mutagenesis and screening strategies on a large scale, and with a cost that is not possible in other vertebrate systems, have allowed the generation of zebrafish models for a wide variety of human diseases [133].

The generation of a zebrafish model with a null allele in *ubiad1*, called barolo (*bar*), produced a phenotype with specific cardiovascular failure due to oxidative stress and ROS-mediated cellular damage [97]. Ubiad1 is considered as a nonmitochondrial prenyltransferase responsible for the synthesis of CoQ_10_ in the Golgi membrane compartment in zebrafish [31,97]. As a result of that, the *bar* mutants showed a depletion in the CoQ_10_ levels in the cytosol. The accumulation of ROS reported in the *bar* mutants increased the lipid peroxidation in vascular cells, leading to the cardiovascular oxidative damage characteristic of this model. Moreover, the inhibition of eNOS rescued the oxidative damage previously described, suggesting a specific role of Ubiad1 on the regulation of NO signaling. Additionally, in the same study, they knocked-down *Coq2* expression during zebrafish development. Loss of Coq2, the mitochondrial prenyltransferase involved in CoQ_10_ biosynthesis, leads to oxidative stress, but it does not significantly affect vascular integrity and survival. Comparing both zebrafish models, the authors found that the Coq2-mediated CoQ_10_ production is mainly for mitochondrial respiratory chain function and energy production; whereas Ubiad1-mediated CoQ_10_ production is important for membrane redox signaling and protection from lipid peroxidation [97]. UBIAD1 is also a novel vitamin K_2_ biosynthetic enzyme screened and identified from the human genome database [134]. Another *ubiad1* mutant (*reddish^s587^*) was generated to demonstrate the essential role of the vitamin K_2_ generated by Ubiad1 in the maintenance of the endothelial cell function and overall vascular homeostasis. The treatment with vitamin K_2_ rescued the vascular phenotype of *ubiad1* mutant *reddish^s587^* but not the cardiac phenotype, suggesting that an alternative Ubiad1/vitamin K-independent pathway may participate in the cardiac function [135].

In other series of experiments, morpholino oligonucleotide (MO) knockdown of *Coq6* or *Coq8b* (=*Adck4*) in zebrafish caused apoptosis in the *Coq6* knockdown zebrafish embryos [57] and nephrosis phenotype of periorbital and total body edema in the *Coq8b* knockdown zebrafish embryos [63], mimicking the nephrotic syndrome phenotype of the patients.

In addition to those already characterized models, the Zebrafish Mutation Project seems to have generated different zebrafish models with mutations in Coq genes (https://zmp.buschlab.org/search/coq; accessed on 26 September 2021), although none of these models have been characterized so far.

### 3.2. Mouse Models of CoQ Deficiency

#### 3.2.1. Mouse Models with Spontaneous Mutation

The polyisoprenyl diphosphate synthase is the enzyme responsible for the formation of the isoprenyl side chain of CoQ in mice and humans. It is a heterotetramer composed of two protein subunits. The genes that encode these subunits are designated *Pdss1* and *Pdss2* in mice, and *PDSS1* and *PDSS2* in humans [105,136]. A mutation in *Pdss2* was reported in mice with inherited kidney disease. The mutation arose spontaneously in an inbred strain of mice, which appeared to be due to an autosomal recessive gene, and it was designated ‘kidney disease’ (*kd*) [125,137]. The first clinical manifestations were developed at 10 weeks of age, and it was characterized by increased proteinuria. This was followed over several weeks by excessive drinking, dilute urine, loss of weight, anemia, and death in 5 to 7 months. Histological examination of kidneys at earlier age reveals kidney damage due to mononuclear cell infiltrate and tubular dilatation with proteinaceous casts in cortical areas, which with time expands throughout the entire kidney and leads to renal failure [137]. Initial studies prior to the identification of the genetic defect suggested an autoimmune mechanism as the pathophysiological mechanism behind this syndrome [138,139]. However, this hypothesis was discarded, as it was proved that the immune response was a secondary consequence of the genetic defect of *kd/kd* mice [140]. *Kd/kd* mice developed a typical nephrotic syndrome with the characteristic biochemical perturbations in serum accompanied by albuminuria and visceral epithelial abnormalities, including hyperplasia and podocyte effacement [125,141].

A positional cloning approach demonstrated that the *kd* allele is a missense mutation in *Pdss2* gene (*Pdss2^kd/kd^*), showing that the failure in the coenzyme Q biosynthetic pathway is the cause of a lethal kidney disease in mice [142]. The *kd/kd* mutation (V117M) occurs within the conserved domain I of *Pdss2*, and the levels of CoQ_9_ and CoQ_10_ in kidney homogenates from *Pdss2^kd/kd^* were significantly lower than in control mice [105]. The *Pdss2* mutant mice manifested widespread CoQ deficiency and abnormalities in the mitochondrial respiratory chain. However, other parameters such as ROS production, oxidative stress, mitochondrial DNA depletion, and citrate synthase activity, an index of mitochondrial mass, appeared only in affected organs. Those data suggested that kidney-specific loss of mitochondria triggered by oxidative stress may be the cause of renal failure in *Pdss2^kd/kd^* mice [98]. Additionally, the impairment of the sulfide oxidation pathway induced by decreased levels of CoQ has been proposed as a pathophysiological mechanism under this syndrome. The disruption in the sulfide oxidation pathway caused the following: (1) accumulation of sulfides due to the decreased levels of SQOR and downstream enzymes, (2) low levels of plasma and urine thiosulfate, (3) the inhibition of short-chain acyl-CoA dehydrogenase, and (4) the decreased levels of glutathione, which may partially avoid the scavenge of hydrogen peroxide, thus contributing to oxidative damage and structural and functional alteration of the renal glomerulus [15,99].

Oral supplementation with CoQ_10_ induces a rescue of proteinuria and interstitial nephritis in the *Pdss2^kd/kd^* mutant mice through the normalization of different CoQ functions [100,101]. Additional studies demonstrated that probucol had a more powerful health improvement than high-dose CoQ_10_ supplementation and was able to restore CoQ_9_ content in the kidneys of *Pdss2^kd/kd^* mice [102]. In addition, rapamycin administration was able to reduce the proteinuria in *Pdss2^kd/kd^* mice [103], and the treatment with GDC-0879, a Braf/Mapk-targeting compound, ameliorated renal disease in *Pdss2^kd/kd^* mice, which was probably through a mechanism that involves the activity of GPx4 [104].

#### 3.2.2. Conditional Knockout Mouse Models

To further evaluate the nephrotic phenotype associated with CoQ deficiency, tissue-specific conditional *Pdss2* knockout (KO) mice were generated [105,125]. The deletion of *Pdss2* was targeted to renal glomerular podocytes in *Podocin/cre*,*Pdss2^loxP/loxP^*, renal tubular epithelium and hepatocytes in *PEPCK/cre,Pdss2^loxP/loxP^*, monocytes in *LysM/cre,Pdss2^loxP/loxP^*, and hepatocytes in *Alb/cre,Pdss2^loxP/loxP^*. Interestingly, the kidney disease phenotype appeared only in *Podocin/cre*,*Pdss2^loxP/loxP^* KO mice but not in the other conditional KO, as estimated by albuminuria and morphological evidence of nephritis (dilated tubules and extensive interstitial infiltration). These data suggested that the renal glomerular podocytes are particularly sensitive to the dysfunction of PDSS2. In fact, knocking out the *Pdss2* gene in podocytes resulted in a more severe phenotype than that observed in *Pdss2^kd/kd^* mice, indicating that the product of the missense allele in *Pdss2^kd/kd^* has some residual activity. The liver-conditional *Alb/cre,Pdss2^loxP/loxP^* KO mice did not show detectable levels of CoQ_9_ in the liver, leading to impaired mitochondrial respiration and altered intermediary metabolism as demonstrated by transcriptional profiling and amino acid quantitation [105]. Nevertheless, *Alb/cre,Pdss2^loxP/loxP^* KO mice did not develop any symptoms of disease.

In addition to kidney, cerebellum is one of the most often affected organs in CoQ deficiency [143], and cerebellar atrophy has been diagnosed in many infants with this disease. In order to analyze the cerebellum defect in CoQ deficiency, a *Pdss2* conditional KO (*Pax2/cre,Pdss2^f/-^*) was generated crossing a previously generated *Pdss2* floxed mouse (*Pdss2^f/f^*) with a *Pax2-cre* deleter mouse, in which Cre recombinase is expressed in the hindbrain region at E9.5 and influences many cells in the cerebellum at birth [106]. *Pax2/cre,Pdss2^f/-^* died within the first 36 h of life and suffered from cerebellar hypoplasia and cellular disorganization in the cerebellum, which mimics the cerebellum atrophy commonly observed in CoQ-deficient infants. This macroscopic observation was accompanied by inhibition of cell migration and cell proliferation as well as increased ectopic apoptosis in the cerebellum of *Pdss2* KO embryos [106]. A different specific deleter mouse *Pcp2-cre* [144] was also crossed with a *Pdss2^f/f^* mouse to generate another tissue-specific KO mouse (*Pcp2/cre,Pdss2^f/−^*). In this last *Pdss2* mutant model, the mutation was developed only in cerebellum Purkinje cells after birth. *Pcp2/cre,Pdss2^f/−^* was clinically healthy and with normal behavior until 4.5 months. From this time, an ataxic phenotype at old age was manifested due to the progressive decrease of Purkinje cells and initiation of diffusive neuron death by apoptosis, showing at 9.5-month loss in motor coordination and incapability of maintaining body balance on a rod. Thus, this last conditional KO mouse model may be a better model to study the cerebellar defects linked to CoQ deficiency in adulthood [106].

As steroid-resistant nephrotic syndrome (SRNS) has been linked to mutations in several genes encoding CoQ biosynthetic enzymes [52,54,57,63], a podocyte-specific *Coq6* KO mouse model was generated by crossing Podocin-*cre* mice with *Coq6^loxP/loxP^* mice in which two *loxP* sites surround exon 6 in the *Coq6* gene. The COQ6 protein is an hydroxylase that seems to catalyze two uncharacterized hydroxylation steps in CoQ biosynthesis [136]. The *Coq6* KO model is referred as *Nphs2/cre,Coq6^loxP/loxP^* KO mice, also *Coq6^podKO^* [107]. *Coq6^podKO^* mice appeared to be normal in development and body condition until the age of 5 months, when they deteriorated, gradually developing progressive glomerular sclerosis and proteinuria. They became moribund at 10 months of age with advanced decline of renal function. Interestingly, the administration of 2,4-dihidroxibenzoate acid (2,4-diHB), a CoQ_10_ precursor analogue, to 5-month-old *Coq6^podKO^* mice significantly protected from disease progression, and the survival was comparable to that of control mice. With the 2,4-diHB treatment, proteinuria and renal histology improved dramatically in treated *Coq6^podKO^* mice compared with untreated *Coq6^podKO^* mice [107]. However, the therapeutic mechanisms of 2,4-diHB in this model are unclear, and the levels of CoQ after the treatment were not reported.

As mutations in *ADCK4 (COQ8B)* have been also reported in patients with SRNS, a similar study was carried out to evaluate the role of *Adck4* in kidney function. For that purpose, a podocyte-specific *Adck4* KO mouse was generated. Known as *Adck4^tm1d^* or *Nphs2/cre, Adck4^flox/flox^* (also *Adck4^ΔPodocyte^*) was generated by crossing the *Nphs2-cre* mouse with the *Adck4^flox/flox^* mouse in which two *loxP* sites surround exons 5 and 6 in the *Adck4* gene. *Adck4^ΔPodocyte^* developed an increased in adult morbidity, mortality, and weight loss with progressive albuminuria and renal structural abnormalities (focal segmental glomerulosclerosis with extensive interstitial fibrosis and tubular atrophy) and functional decline. Treatment of 3-month-old *Adck4* KO mice with 2,4-diHB prevented the development of renal pathology and reversed mitochondrial dysfunction. Again, the CoQ levels after 2,4-diHB therapy were not reported, so the therapeutic mechanisms were unknown. Nevertheless, these data suggest that ADCK4, an uncharacterized mitochondrial protein with no enzymatic activity directly involved in the CoQ biosynthetic pathway, must be a protein component in CoQ biosynthesis [108]. The rescue in the phenotypes of the *Coq6^podKO^* and *Adck4^ΔPodocyte^* KO mice after the treatment with 2,4-diHB suggests a potential treatment strategy for nephrotic syndrome resulting from *COQ6* and *ADCK4* mutations.

Separately, the *Coq7*(=*Mclk1*)*^liver-KO^* mouse was generated. This model carried a liver-specific KO mutation in the *Coq7* gene. The quinone measurements in the liver of *Coq7^liver-KO^* showed that CoQ_9_ levels were decreased by 85% and DMQ_9_ was substantially accumulated, which is consistent with a dysfunctional COQ7 activity. This decrease in CoQ in hepatocytes caused only a mild reduction of the mitochondrial respiratory chain function and no gross abnormalities, suggesting a nonlinear dependence of mitochondrial respiratory capacity on CoQ content. In addition, these results suggest that very little CoQ is required in this process. Moreover, DMQ seems not to interfere with CoQ-mediated mitochondrial electron transport in the liver. The supplementation with CoQ_10_ to *Coq7^liver-KO^* increased the CoQ_10_ content in the liver and partially rescued the electron transport deficit in this tissue [109]. Later on, a global conditional KO mouse model for the *Coq7* gene was generated by expressing tamoxifen TM-dependent CreER^T2^ transgene. Following the induction of *Coq7* KO by TM at two months of age, adult-onset global *Coq7* KO (aog*Coq7*) animals gradually accumulated DMQ and lost CoQ, leading to mitochondrial dysfunction and shortened lifespan. Interestingly, during the time that *Coq7* KO mice started to die, the levels of CoQ were very low in the heart. However, cardiac function was not compromised, suggesting that this tissue has a CoQ reservoir. Dietary CoQ_10_ was ineffective in rescuing the phenotype of *Coq7* KO mouse mice, but the supplementation with 2,4-diHB produced a partial restoration of mitochondrial respiration and led to a marked increase in their survival [110]. In this particular case, the therapeutic mechanism was attributed to the increase in CoQ and decrease in DMQ, as 2,4-diHB is able to bypass defects in COQ7. This result highlighted the important clinical implications of the treatment with 2,4-diHB also in *COQ7* mutations. Furthermore, a late-onset treatment with 2,4-diHB 6 months after TM induction, a moment in which the untreated KO mice show apparent phenotypes, was also able to increase the survival the *Coq7* KO mice [110].

#### 3.2.3. Constitutive Knockout and Knock-In Mouse Models

In addition to the mouse models with spontaneous mutation and the conditional KO mouse models of CoQ deficiency previously described, some constitutive KO and knock-in (KI) models have been generated to date.

The KO for *Pdss2* (B6.*Ȥp3/cre,Pdss2^loxP/loxP^*) was embryonically lethal, with no homozygous embryos surviving beyond 10.5 days of gestation [105], which is consistent with the hypothesis that ubiquinone is essential for mouse embryonic development [106,112]. Similar results were obtained with the constitutive KO mouse for *Coq3* [111]. COQ3 is an O-methyltransferase responsible for the first and last O-methyltransferase steps in CoQ biosynthesis [145]. *Coq3^−/−^* resulted in embryonic lethality, as no *Coq3* homozygous mice were obtained from heterozygous crosses [111]. However, *Coq3* heterozygous mouse (*Coq3^+/−^*) showed normal lifespan and no changes in CoQ levels in pure mitochondria isolated from 3-month-old *Coq3^+/−^* livers [111].

Similar to *Pdss2* and *Coq3* KO, two independently generated KO models of *Coq7* showed embryonic lethality [112,113]. In one case, the *Coq7^−/−^* embryos failed to survive beyond E10.5, exhibiting small-sized body and delayed neural development. These results suggest that COQ7-deficient embryos are able to develop until E8 with the energy generated from anaerobic glycolysis, but they are unable to survive due to the lack of aerobic glycolysis required for further embryogenesis in mice. Electron microscopic analysis showed a loss of organized neuroepithelial structures in *Coq7^−/−^* embryos and enlarged mitochondria with vesicular cristae and enlarged lysosomes filled with disrupted membranes, demonstrating that COQ7 is essential for neurogenesis and for the maintenance of mitochondrial integrity [113]. In the other study, the *Coq7^−/−^* embryos showed a developmental delay that was evident by E9.5, and all *Coq7^−/−^* embryos detected were completely resorbed by E13.5 [112]. Consistent again with a dysfunctional COQ7 protein, the embryos of both KO *Coq7* mice showed reduced CoQ levels and accumulation of DMQ, highlighting the crucial role of COQ7 in the CoQ metabolism [112,113].

On the other hand, heterozygous *Coq7^+/−^* mice were viable and fertile with no obvious anatomical or behavioral defects [112]. The amounts of CoQ_9_ and CoQ_10_ were similar in wild-type and heterozygous embryos. Young *Coq7^+/−^* mutants showed reduced mitochondrial oxygen consumption, reduced electron transport, reduced mitochondrial ATP synthesis, reduced mitochondrial and overall ATP levels, and reduced whole-animal oxygen consumption [146]. Moreover, it was discovered that in the livers of very old *Coq7^+/−^* mutants, a phenomenon of loss-of-heterozygosity took place because there were areas of complete loss of expression of *Coq7*. That was followed by a decrease in the CoQ levels in the livers of very old *Coq7^+/−^* mutants [111,146]. Furthermore, it has been reported that *Coq7^+/−^* mice present a unique mitochondrial CoQ profile that was characterized by decreased CoQ levels in the inner mitochondrial membrane coupled with higher CoQ levels in the outer mitochondrial membrane. The low levels of CoQ in the inner membrane could increase oxidative stress by partially inhibiting the electron transport chain. At the same time, similar amounts of total CoQ were detected in the mitochondrial, peroxisomal, and plasma membrane fractions from *Coq7^+/−^* and control mice livers at three months of age. Dietary supplementation of *Coq7^+/−^* mice with CoQ_10_ normalized the CoQ levels in the inner mitochondrial membrane and in the outer mitochondrial membrane as well as the respiratory chain dysfunction [111]. Interestingly, *Coq7^+/−^* mice showed an increase in lifespan up to 31%, as it was reported in the mutational inactivation of its orthologue *clk-1* in *C. elegans*, with significantly lower levels of DNA damage [114]. The increase in the lifespan in *Coq7^+/−^* mice has been associated with an early hepatic mitochondrial dysfunction, which induces a protective physiological response called mitohormesis [114]. The mechanisms underlying the increased lifespan in *Coq7^+/−^* mice include the following:

(1) Despite the early mitochondrial dysfunction, the function of *Coq7^+/−^* mitochondria declines less rapidly with age than that of the wild type, and there is a slower accumulation of global oxidative biomarkers of aging in these mutants [147].

(2) The altered mitochondrial phenotype of *Coq7^+/−^* mutants enhanced immune reaction to infection by improving basal and stimulated expression of HIF-1α in liver and macrophages, in association with elevated expression of inflammatory cytokines [148].

(3) *Coq7^+/−^* mutants have enhanced resistance to neurological damage after ischemia and reperfusion [149].

Finally, a recent study in *Coq7^+/−^* mutants suggested that *Coq7* regulates microglial metabolic reprogramming participating in neuroinflammation and dopaminergic cell death [150].

In contrast with the *Coq* gene KOs previously mentioned, the lack of COQ8A in mice (*Coq8a^−/−^*) resulted in a mild phenotype with progressive cerebellar ataxia, mild exercise intolerance, and moderate CoQ deficiency, recapitulating the more frequent features of autosomal-recessive cerebellar ataxia type 2 (ARCA2, the most frequent form of hereditary CoQ deficiency in humans) [115]. The pathophysiology of the disease was linked to dysfunctional cerebellar Purkinje cells, defective skeletal muscle, and disruption of complex Q. *Coq8a^−/−^* mice develop mild, tissue-specific CoQ deficiency. The levels of CoQ were low in skeletal muscle of *Coq8a^−/−^* mice, while normal CoQ levels were observed in whole *Coq8a^−/−^* cerebellum. These data suggest that CoQ deficiency may specifically affect the Purkinje cells of the cerebellum, although the specific measurement of CoQ in those cells was not performed. Additionally, complex Q proteins were deficient across multiple *Coq8a^−/−^* tissues, showing that COQ8A participates in the stability of complex Q in mammals. Moreover, *Coq8a^−/−^* Purkinje cells displayed Golgi morphology defects, but normal mitochondria, providing a model system to study CoQ production across different organelles, cells, and tissues [115].

Another mouse model with intriguing results is the KO model for *Coq9* (*Coq9^Q95X^*), which had normal development, in contrast with the embryonic lethality described in *Pdss2*, *Coq3,* and *Coq7* KO mice. The *Coq9^Q95X^* mouse model was generated by the Wellcome Trust Sanger Institute from ES cell clone EPD0112_2_A09 obtained from the supported KOMP Repository (www.komp.org; accessed on 26 September 2021). Lack of the COQ9 protein caused moderate CoQ deficiency; i.e., the cerebrum, cerebellum, and heart showed around 50% residual CoQ_9_ levels, while the kidney and skeletal muscle had 30% of residual CoQ_9_ levels compared with wild-type mice. This led to a reduction in CI+III activity and mitochondrial respiration in skeletal muscle and late-onset mild mitochondrial myopathy with exercise intolerance in female mice. The COQ9 protein is needed for the stability and activity of COQ7 in the CoQ biosynthetic pathway, so the brain, kidneys, and muscles of *Coq9^Q95X^* mice also showed a decrease in the levels of COQ7 [116]. As in other animal models with a decrease in the COQ7 levels [146,151], *Coq9^Q95X^* mice showed an increase in lifespan, especially in males because they lived on average 15% longer than their wild-type littermates. The increase in lifespan in *Coq9^Q95X^* mice is accompanied by a reduction in the animals’ body weight [117]. However, unlike *Coq7^+/−^* mice, the liver of *Coq9^Q95X^* mice showed normal levels of CoQ levels; consequently, mitochondrial function was also normal in this tissue. Therefore, unlike *Coq7^+/−^* mice, the hepatic mitochondrial dysfunction and increased oxidative stress induced by subphysiological levels of CoQ biosynthetic proteins are not the cause of the increased lifespan in *Coq9^Q95X^* mice. These effects could be mediated by mechanisms initiated in the mitochondria from other tissues or by other unknown mechanisms [117].

To better understand the pathomechanisms of primary CoQ deficiency due to a mutation in the *Coq9* gene, a knock-in mouse model carrying a homozygous mutation in *Coq9* gene was generated (*Coq9^R239X^*) [45]. The *Coq9^R239X^* mice encoded the mutation CGT > TGA (R239X) within exon 7 in the mouse genome, which is a homologue to the human R244X mutation described in a patient with CoQ_10_ deficiency [64]. In contrast with the *Coq9^Q95X^* mouse, the *Coq9^R239X^* model manifested severe CoQ deficiency; i.e., the CoQ_9_ levels were 15–20% compared to wild-type animals. The deficit in CoQ induced a decrease in mitochondrial respiration, particularly in the brain and kidneys. These effects lead to neuronal death and demyelination with spongiosis and astrogliosis in the brain of *Coq9^R239X^* mice, leading to a premature death [45,116]. In tissues from *Coq9^R239X^* mice, a truncated version of the COQ9 protein was detected, leading to a reduction in the COQ7 protein and, as a consequence, a widespread CoQ deficiency and accumulation of DMQ [45,116]. The truncated version of COQ9 protein in *Coq9^R239X^* mice destabilizes the complex Q and produces a severe phenotype associated with fatal encephalomyopathy. Therefore, the stability of the complex Q clearly influences the CoQ biosynthesis rate and, consequently, the degree of the severity of CoQ deficiency and the development of tissue-specific phenotypes [116]. Moreover, the disruption of mitochondrial sulfide oxidation has been identified as one of the pathomechanisms associated to CoQ deficiency in both *Coq9^Q95X^* and *Coq9^R239X^* mice [118], similarly to the *Pdss2^kd/kd^* mice [15]. Specifically, severe CoQ deficiency caused a remarkable reduction in SQOR levels and activity, and this deficit induced changes in the mitochondrial sulfide metabolism. In the brain of *Coq9^R239X^* mice, the low SQOR levels produced an increase in downstream enzymes as well as adjustments in the levels of thiols. As a result, the biosynthetic pathways of glutamate, serotonin, and catecholamines were altered in the cerebrum, and the blood pressure was reduced [118]. Additionally, CoQ deficiency in *Coq9^R239X^* mice also affected the sulfide biosynthetic pathway (also known as the transsulfuration pathway), independently of the availability of sulfur amino acids. In the kidneys of *Coq9^R239X^* mice, the levels of cystathione-β-synthase, an enzyme of the transsulfuration pathway, marginally increased compared with the control mice, and this variation was maintained under supplementation with N-acetylcysteine or a sulfur amino acids restriction diet, displaying the tight regulation between the biosynthetic and catabolic pathways of sulfide metabolism [119].

Oral supplementation with ubiquinone-10, the oxidized form of CoQ_10_, had limited efficacy in the treatment of the *Coq9^R239X^* mouse model due to its poor absorption and bioavailability. The supplementation with ubiquinol-10, the reduced form of ubiquinone-10, provided better therapeutic results. It increased the levels of CoQ_10_ in tissue homogenates and cerebral mitochondria, which results in an increase in CoQ-dependent respiratory chain activities, reduction in the spongiosis, astrogliosis, and oxidative stress in different brain areas, and an increase in body weight. These data suggest that water-soluble formulations of ubiquinol-10 are a better option than ubiquinone-10 for the treatment of primary CoQ_10_ deficiency [120]. Looking for alternative strategies to the classical exogenous CoQ_10_ supplementation with the aim of increasing the endogenous CoQ biosynthesis, the therapeutic potential of 2,4-diHB (β-resorcylic acid or β-RA in the study) was also tested in *Coq9^R239X^* mice [121,152]. Remarkably, the treatment in *Coq9^R239X^* mice increased the lifespan to values close to the lifespan in wild-type mice. Moreover, while the maximal survival of *Coq9^R239X^* mice treated with ubiquinol-10 was 17 months of age, the lifespan achieved by 2,4-diHB reached a maximum of 25 months of age [121]. 2,4-diHB supplementation rescued the phenotype of *Coq9^R239X^* mice, as shown by the reduction in the histopathological signs of the encephalopathy, i.e., the spongiosis and reactive astrogliosis. Those effects were mainly linked to the decrease in the levels DMQ_9_ and the increase in mitochondrial bioenergetics in peripheral tissues. However, even if the CoQ biosynthesis or the mitochondrial function did not change in the brain after the therapy, the *Coq9^R239X^* mice showed an almost absence of cerebral vacuoles and reactive astrocytes after the 2,4-diHB treatment. The authors suggested the hypothesis of a possible tissue–brain cross-talk, but further studies are needed to reveal how the peripheral tissues communicate with the brain in this situation [121,152].

Two other therapeutic strategies have been tested using the *Coq9^R239X^* mouse model or cells from this mouse model. Firstly, the use of a lentiviral vector (CCoq9WP) allowed the ectopi over-expression of *Coq9* in mouse embryonic fibroblasts (MEFs) and hematopoietic progenitor cells (HPC), leading to the restore of the CoQ biosynthetic pathway and mitochondrial function [122]. Secondly, researchers tested the therapeutic effects of rapamycin administration in *Coq9^R239X^* mice [153] based on other studies that had demonstrated the therapeutic benefits of rapamycin therapy in a few mouse models of mitochondrial diseases [154,155]. Neither a low nor a high dose of rapamycin were able to increase the mitochondrial bioenergetics, to reduce the spongiosis and reactive astrogliosis, and to rescue the phenotypic characteristics of *Coq9^R239X^* mice, resulting in the lack of efficacy for increasing the survival [153].

Recently, a heterozygous *Adck2* KO mouse model (*Adck2^+/−^*) has been generated to clarify the role of ADCK2 in CoQ biosynthesis and to explain the pathological mechanisms involved in the disease caused by a mutation in this gene. While *Adck2^−/−^* mice are embryonically lethal, *Adck2^+/−^* mice partially recapitulated the phenotype of a human patient characterized by an adult-onset myopathy due to CoQ deficiency and an overall defect in mitochondrial lipid metabolism. The quinones determination showed a significant decrease in CoQ_9_ and CoQ_10_ levels in the skeletal muscle of *Adck2^+/−^*, leading to a decrease in mitochondrial complex I+III and II+III activities and oxygen consumption rate. These results suggest that ADCK2 deficiency impaired the normal mitochondrial respiratory chain function in skeletal muscle, ultimately leading to mitochondrial dysfunction. Further transcriptomics and metabolomics analysis supported the hypothesis that ADCK2 plays an important role in the energy homeostasis in skeletal muscle. Based on these data, it was proposed that ADCK2 is involved in the regulation of mitochondrial fatty acid β-oxidation and CoQ levels in skeletal muscle. The supplementation with CoQ_10_ partially rescued the phenotype of *Adck2^+/−^* mice, indicating the possibility of improving some of the defects stemming from CoQ deficiency due to *Adck2* mutation [123].

A mouse model of secondary CoQ deficiency of genetic origin is also available [124]. PARL is a protease located in the inner mitochondrial membrane with relevant but unclear physiological roles. However, it is related to different prevalent human diseases, including cancer and neurodegenerative diseases, highlighting its biological significance in pathological conditions [156]. *Parl^−/−^* mice developed a necrotizing encephalomyopathy closely resembling Leigh syndrome. Mitochondria from the brain of *Parl^−/−^* mice showed progressive structural alterations and early deficiencies of complex III (CIII) and CoQ, resulting in a disruption of the mitochondrial calcium metabolism. A mitochondrial proteome analysis in the brain of the PARL-deficient mice showed downregulation of the CIII-regulating protein TTC19, several proteins required for CoQ biosynthesis, and SQOR. The disruption in the COQ4 levels was detected in *Parl^−/−^* brains already at one week of age. Subsequently, a downregulation of COQ3, COQ5, COQ6, COQ7, COQ9, and SQOR was also observed. As a conclusion, the data indicated that the stabilization of TTC19 and the following regulation of CIII activity is mediated by PARL as well as the maintenance of COQ4 expression in the brain, supporting CoQ biosynthesis in this organ. Nevertheless, further studies are needed to explain the connection between the disruption in PARL and in COQ4 and the resulting CoQ deficiency in the brain [124].

## 4. Conclusions and Perspectives

Primary CoQ deficiency is caused by mutations in any of the genes involved in the CoQ biosynthetic pathway. Clinically, CoQ deficiency is a heterogeneous disease in humans, which is linked to five major phenotypes and some other clinical symptoms. The clinical heterogeneity and the variety of functions associated with CoQ complicate the description of the pathomechanisms underlying this mitochondrial disease as well as the development of new therapies. For that reason, several animal models with CoQ deficiency have been generated, providing very useful insights, such as: (1) they have confirmed the involvement of the target proteins in CoQ biosynthesis in different species, indicating the evolutionary conservation of the pathway; (2) they have clearly shown the importance of CoQ during development and its role in the mitochondrial respiratory chain, sulfide metabolism, and pyrimidine metabolism; (3) they have revealed that the genotype–phenotype association, the bioenergetics defect, the increased oxidative stress, and the disruption of sulfide and pyrimidine metabolism are key disease pathomechanisms that may explain, at least in part, the tissue specificities and the clinical heterogeneity; (4) they have shown the limitation of the exogenous CoQ_10_ therapy, especially in some particular phenotypes; and (5) they have opened promising therapeutic strategies based on the supplementation with 4-HB analogs. However, some functional roles of CoQ have not been evaluated yet in the context of CoQ deficiency. In addition, the therapeutic mechanisms of 2,4-diHB in some models seem to be independent of the CoQ biosynthetic pathway, increasing the potential relevance of this compound in the clinical practice. Furthermore, the development of other animal models is required to cover the study of this complex syndrome and the different functions of CoQ, and to elucidate the synthesis of CoQ in mammals. This is especially important, for example, in the case of CoQ deficiency due to mutations in the *COQ2* gene. In vitro studies have demonstrated the therapeutic potential of the natural precursor of CoQ, 4-hydroxybenzoic acid (4-HB) [157], but there is no available mouse model with mutation in *Coq2* to extend the study, limiting the potential translation of this treatment into the clinic.

## Figures and Tables

**Figure 1 antioxidants-10-01687-f001:**
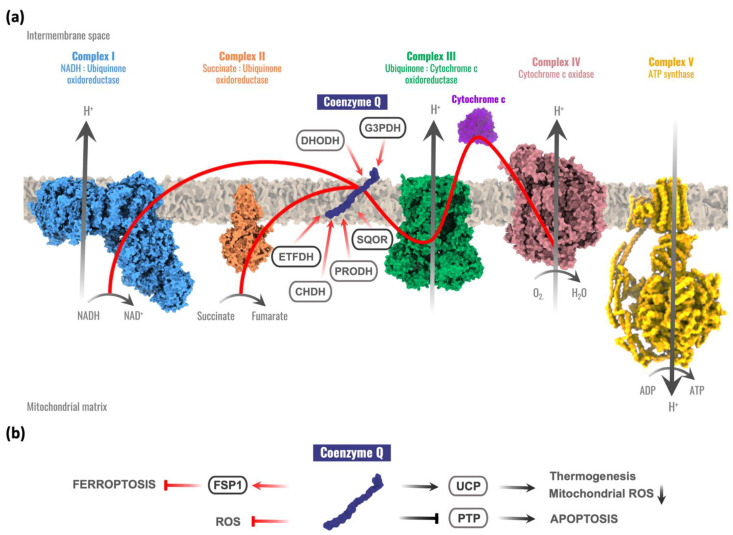
Coenzyme Q functions in the cell. (**a**) The role of coenzyme Q (CoQ) in the mitochondrial respiratory chain and its relationships with other mitochondrial enzymes. Red arrows represent electron flow. CoQ accepts electrons from complex I and complex II, sulfide:quinone oxidoreductase (SQOR), proline dehydrogenase and proline dehydrogenase 2 (PRODH), choline dehydrogenase (CHDH), mitochondrial glycerol-3-phosphate dehydrogenase (G3PDH), dihydroorotate dehydrogenase (DHOH), and electron transport flavoprotein dehydrogenase (ETFDH). (**b**) CoQ extramitochondrial functions. Red arrows represent electron flow. FSP1 = Ferroptosis Suppressor Protein 1; UCP = Uncoupling Protein; PTP = Permeability Transition Pore; ROS = Reactive Oxygen Species.

**Figure 2 antioxidants-10-01687-f002:**
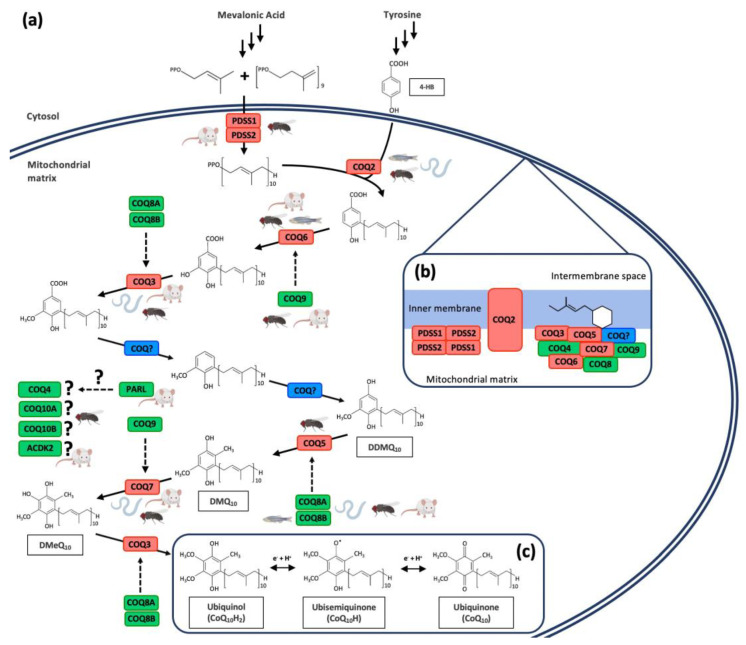
Coenzyme Q_10_ biosynthetic pathway and CoQ_10_ redox cycle. (**a**) Schematic model of human CoQ_10_ biosynthetic pathway. The red color indicates proteins with enzymatic activity. The green color indicates proteins with regulatory function. The blue color shows currently unidentified enzymes. Figures of mouse, *C. elegans*, *D. melanogaster*, and zebrafish illustrate animal models with mutation in each specific protein. (**b**) Model of human CoQ_10_ biosynthetic complex, containing at least COQ3–COQ9 and lipids, such as CoQ itself. (**c**) Chemical structure of coenzyme Q_10_ and its redox cycle. 4-HB = 4-hydroxybenzoic Acid; DDMQ = demethoxy-demethyl-ubiquinone; DMQ = demethoxyubiquinone; DMeQ = demethylubiquinone.

**Table 1 antioxidants-10-01687-t001:** Overview of the animal models of CoQ deficiency mentioned, remarking their contribution to expand the knowledge about the CoQ biosynthetic pathway and functions, and the pathomechanisms and treatments related to CoQ deficiency.

Model	Strain	Phenotype	Coenzyme Q		CoQ Deficiencies		Ref
			Biosynthesis	Functions	Pathological Mechanisms	Treatments	
*Drosophila melanogaster*	*qless* mutant	Nervous system failure			Caspase-dependent apoptosis in neurons	CoQ_4_, CoQ_9_ or CoQ_10_ rescue the phenotype.	[78]
	*sbo* mutant	Small larvae phenotype Controversial extended lifespan		CoQ is important in the early stages of development and fertility.	More susceptible to bacterial and fungal infections	CoQ_10_ partially rescue the phenotype.	[79,80]
	*Coq2* mutant	Renal failure Larval lethality		CoQ is important in the early stages of development.	ROS accumulation	Glutathione and vanillic acid rescued the phenotype.	[81,82]
	*Coq3* mutant	Larval lethality	Existence of complex Q	CoQ is important in the early stages of development.	DMQ accumulation		[82,83]
	*Coq5* mutant	Larval lethality		CoQ is important in the early stages of development.			[82,83]
	*Coq6* mutant		Existence of complex Q		DMQ accumulation		[82,83]
	*Coq7* mutant	Larval lethality	Existence of complex Q	CoQ is important in the early stages of development.	DMQ accumulation		[82,83]
	*Coq8* mutant	Larval lethality		CoQ is important in the early stages of development.			[82,83]
	*Coq9* mutant	Larval lethality	Existence of complex Q	CoQ is important in the early stages of development.	DMQ accumulation		[82,83]
	*Coq10* mutant	Larval lethality		CoQ is important in the early stages of development.			[82,83]
*Caernorhabditis elegans*	*Mev-1* mutant	Shortened lifespan		CoQ is an important link between lifespan and oxidative stress.	ROS accumulation. Altered reduced and oxidized forms of CoQ	CoQ_10_ rescued the phenotype.	[84,85,86]
	*Clk-1* mutant	Extended lifespan	COQ7 has catalytic activity.	CoQ_10_ has the highest antioxidant properties.	Accumulation of DMQ_9_. Lower production of ROS. Inhibition of complex I by DMQ_9_	CoQ_10_ restored the phenotype.	[87,88,89,90,91,92]
	*Coq-1* mutant	Larval lethality Muscle failure Nervous system failure		CoQ is important in the early stages of development and fertility.	Degeneration of GABA neurons	CoQ_10_ or feeding worms with bacteria containing its own CoQ_8_ did not improved the phenotype	[92,93,94]
	*Coq-2* mutant	Larval lethality Muscle failure Nervous system failure		CoQ is important in the early stages of development and fertility.	Degeneration of GABA neurons	CoQ_10_ or feeding worms with bacteria containing its own CoQ_8_ did not improved the phenotype	[92,93,94]
	*Coq-3* mutant	Larval lethality Nervous system failure		CoQ is important in the early stages of development and fertility.	Degeneration of GABA neurons	Extra-chromosomal array containing the own *C. elegans coq-3* gene rescue the phenotype.	[92,94,95]
	*Coq8* mutant	Larval lethality		CoQ is important in the early stages of development and fertility.		CoQ-rich diet increased its lifespan	[93,96]
*Danio rerio* (zebrafish)	*ubiad 1 (bar)* mutant	Cardiovascular failure	UBIAD1 participates in non-mitochondrial CoQ biosynthesis.	UBIAD1-mediated CoQ_10_ production is important for mem-brane redox signaling and protection from lipid peroxidation.	ROS accumulation.		[97]
	*Coq2* mutant	Normal		COQ2-mediated CoQ_10_ production is for mitochondrial respiratory chain function and energy production.	ROS accumulation.		[97]
	*Coq6* mutant				Apoptosis increase.		[57]
	*Coq8b* mutant	Renal failure					[63]
*Mus Musculus*	*Pdss2^kd/kd^*	Renal failure	Pdss2 participates in the biosynthesis of CoQ.		Disruption of sulfide oxidation pathway. Inhibition of short-chain acyl-CoA dehy-drogenase. Glutathione depletion.	CoQ_10_ rescued proteinuria and intersti-tial nephritis. Probucol had a more powerful health improvement than high-dose CoQ_10_. Rapamycin reduced the proteinuria. Treatment with GDC0879 ameliorated kidney disease.	[98,99,100,101,102,103,104]
	*Podocin/cre*, *Pdss2^loxP/loxP^*	Nephrotic syndrome			Renal glomerular podocytes display the greatest sensitivity to *Pdss2* impairment		[105]
	*Pdss2f/-; Pax2-cre*	Cerebellar ataxia			Defect in cell migration, cell proliferation, and increased apoptosis in cerebellum		[106]
	*Pdss2^f/−^; Pcp2-cre*	Cerebellar ataxia			Ataxia at old age. Progressive decreased of Purkinje cells and neuron death by apoptosis		[106]
	*Coq6^podKO^*	Nephrotic syndrome				2,4-diHB rescued the phenotype.	[107]
	*Adck4^ΔPodocyte^*	Nephrotic syndrome	ADCK4 is a protein component in CoQ biosynthesis with no catalytic activity.			2,4-diHB rescued the phenotype.	[108]
	*Coq7^liver-KO^*	Hepatic failure		A small amount of CoQ is required in the respiratory chain function in liver.	Accumulation of DMQ_9_. DMQ seems not to interfere with CoQ-mediated mitochondrial electron transport in the liver	CoQ_10_ partially rescued the phenotype.	[109]
	aog*Coq7* (*Coq7* KO by TM)	Early death, but unclear phenotype		Energy-demanding tissues are not highly susceptible to CoQ deficit and defective mitochondrial energy metabolism.	Accumulation of DMQ_9_	CoQ_10_ was ineffective. 2,4-diHB partially rescued the phenotype.	[110]
	*Pdss2^−/−^*	Embryonic lethality		CoQ is important in the early stages of development.			[105]
	*Coq3^−/−^*	Embryonic lethality		CoQ is important in the early stages of development.			[111]
	*Coq7^−/−^*	Embryonic lethality		CoQ is important in the early stages of development.	Accumulation of DMQ		[112,113]
	*Coq7^+/−^*	Extended lifespan			Decreased CoQ levels in the inner mitochondrial membrane coupled with higher CoQ levels in the outer mitochondrial membrane. Early hepatic mitochondrial dysfunction, which induces a protective physiological response (mitohormesis)	CoQ_10_ normalized the CoQ levels.	[111,114]
	*Coq8a^−/−^*	Cerebellar ataxia	COQ8a participates in the stability of complex Q.		Purkinje cells displayed Golgi morphology defects, but normal mitochondria. Model to study CoQ production across different organelles, cells, and tissues		[115]
	*Coq9^Q95X^*	Extended lifespan	COQ9 is needed for the stability and activity of Coq7.		Disruption of mitochondrial sulfide oxidation pathway		[116,117]
	*Coq9^R239X^*	Encephalopathy	COQ9 is needed for the stability and activity of COQ7.		Accumulation of DMQ. Disruption of mitochondrial sulfide oxidation pathway	Ubiquinone-10 had limited efficacy. Ubiquinol-10 provided better therapeutic outcomes. 2,4-diHB full rescued the phenotype. Rapamycin has no efficacy.	[45,116,118,119,120,121,122]
	*Adck2^+/−^*	Myopathy	ADCK2 participates in the control of CoQ levels.		Impairment in lipid metabolism	CoQ partially recovered the phenotype.	[123]
	*Parl^−/−^*	Encephalopathy	Parl is required for efficient CoQ biosynthesis in the brain by stabilization of COQ4 expression.		Downregulation of the CIII-regulating protein TTC19		[124]

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
