# Peer review of "Animal Models of Coenzyme Q Deficiency: Mechanistic and Translational Learnings"

_antioxidants, 2021, doi:10.3390/antiox10111687_

Round 1
Reviewer 1 Report
I carefully read the review with interest. It is well written in easy-to-understand and clear English for general readers and showing useful some historical and many latest references.
I believe many readers are interested in animal models of CoQ deficiency, since the synthetic pathway and physiological functions of CoQ, and the effects of the deficiency on the diseases and aging have not been fully elucidated yet. I believe that this review of variety of animal models of CoQ deficiency is very useful for many researchers for better understanding the pathological features and molecular mechanism of CoQ deficiency and developing effective treatments. Abstract and introduction are reasonable, and I agree with them. Especially Figure 1a and Figure 2 are very clear and useful.
But, I would like to point-out some questions and recommendation.
Major point
- I think Table 1 shows overview of various animal models which are revealed in this review and gives us variable information, but it is complicated and difficult to read smoothly. If possible, please improved it . Could you show the pathological phenotype of each model, for example “renal failure”, “cardiac failure”, “muscle failure”, “extended lifespan” “embryonic lethal” ”normal” and so on.  There are no descriptions except for reference in some animal models (Coq6 mutant, Cop8b mutant in Zebrafish, Coq6podKo in mice, and so on).  
Minor points
- Page 9 line 169, please add the description of mev-1 mutant in the Table1. It has short lifespan and increased ROS and rescued by CoQ10 supplementation.
- Page 14 line 427, I think that “Following induction of Coq7 KO by TM at the two months of age, adult onset global Mclk1KO (aogCoq) animals” is easy to understand.
- The author describes the lifespan extend effects of some animal model. Could you please summarize briefly the relationship between the CoQ and aging in animals and human?
Reviewer 2 Report
I recommend minor abstract revision
